# Theoretical Analysis for Using Pulsed Heating Power in Magnetic Hyperthermia Therapy of Breast Cancer

**DOI:** 10.3390/ijms22168895

**Published:** 2021-08-18

**Authors:** Thanh-Luu Cao, Tuan-Anh Le, Yaser Hadadian, Jungwon Yoon

**Affiliations:** 1School of Integrated Technology, Gwangju Institute of Science and Technology, 123 Cheomdangwagiro, Buk-gu, Gwangju 61005, Korea; caothanhluu1988@gm.gist.ac.kr (T.-L.C.); or letuananh@tlu.edu.vn (T.-A.L.); yaser.hadadian@gmail.com (Y.H.); 2Department of Electrical Engineering, Faulty of Electrical and Electronics Engineering, Thuyloi University, 175 Tay Son, Dong Da, Hanoi 116705, Vietnam

**Keywords:** pulsed heating power, magnetic nanoparticles, fraction of damage, breast cancer, hyperthermia

## Abstract

In magnetic hyperthermia, magnetic nanoparticles (MNPs) are used to generate heat in an alternating magnetic field to destroy cancerous cells. This field can be continuous or pulsed. Although a large amount of research has been devoted to studying the efficiency and side effects of continuous fields, little attention has been paid to the use of pulsed fields. In this simulation study, Fourier’s law and COMSOL software have been utilized to identify the heating power necessary for treating breast cancer under blood flow and metabolism to obtain the optimized condition among the pulsed powers for thermal ablation. The results showed that for small source diameters (not larger than 4 mm), pulsed powers with high duties were more effective than continuous power. Although by increasing the source domain the fraction of damage caused by continuous power reached the damage caused by the pulsed powers, it affected the healthy tissues more (at least two times greater) than the pulsed powers. Pulsed powers with high duty (0.8 and 0.9) showed the optimized condition and the results have been explained based on the Arrhenius equation. Utilizing the pulsed powers for breast cancer treatment can potentially be an efficient approach for treating breast tumors due to requiring lower heating power and minimizing side effects to the healthy tissues.

## 1. Introduction

Breast cancer is the most common malignancy in women aged 50 to 70 years old. The mortality rate due to breast cancer is rather high with a ratio of 1 per 5000 patients worldwide [1,2]. Treatment for breast cancer usually combines surgery with radiation-, hormone- and/or chemotherapy [1]. Hormone- and/or chemotherapy are used for malignancy that has spread beyond the breast (metastases) and lymph nodes. In addition, there two common types of surgery: breast-conserving surgery for only removing the breast lump and a rim of surrounding normal breast tissue, and mastectomy surgery for removing the whole breast including the nipple [1,3,4]. The mortality rate of breast cancer depends on its stages. There are five main stages for breast cancer: stage 0, carcinoma in situ or noninvasive breast cancer; stage I, tumors less than 2 cm; stage II, tumor diameters from 2 to 5 cm with only lymph nodes on the same side of breast cancer; stage III, tumors infiltrating happens in the lymph nodes but without metastases; and stage IV, metastases. The stages I and II are simply cured by surgery with mortality rate less than 50%, while the more complex treatment integrating surgery, chemotherapy and radiotherapy is required in stage IV resulting in mortality rate of more than 50% [1,5,6].

Breast cancer spreading to lymph nodes is usually treated by corresponding levels of radiation therapy of the whole breast and the supraclavicular [5]. Radiation therapy using high-energy radiation such as X-rays, gamma rays, electrons or radioactive material is applied to destroy the tumor left in the breast, chest wall or lymph nodes after the surgery has been performed. Depending on tumor diameter, metastases and lymph nodes, the whole breast is radiated with an added boost to prevent tumor reappearance [2]. However, the conventional therapies result in side effects such as hair loss, weight gain, menopause, osteoporosis, etc. [1]. Several studies have shown the efficacy of hyperthermia for treating cancerous diseases without such side effects [1]. Hyperthermia induces a temperature increase to above 40 °C in the specific time intervals from minutes to hours depending on the type of cancer. The objective of hyperthermia is to cease the growth of tumor cells while minimizing injury to the surrounding healthy tissues. Different types of hyperthermia such as microwave [7,8], radiofrequency waves [9,10], ultrasound [11,12], infrared radiation [13,14], etc. are used in oncology. Hyperthermia integrated with other common therapies such as immunotherapy, chemotherapy and radiotherapy can be used as an adjuvant to reach higher efficiency for preventing locoregional recurrences of breast cancer [15,16,17]. The per session treatment time for breast cancer can vary from 20 to 60 min [18,19,20].

Due to some limitations of traditional hyperthermia, such as the requirement for implanting devices into human body and poor penetration into deep tissues, magnetic particle hyperthermia (MPH) has become an emerging candidate [21]. In this method, magnetic particles, specifically magnetic nanoparticles (MNPs), are injected into the tumor and exposed to an alternating magnetic field to act as a heating source for killing the nearby cancer cells. Due to the interaction between the applied magnetic field and MNPs, energy is dissipated which appears as heat. This energy dissipation, which is proportional to the field parameters, the intrinsic properties of the particles, and their surrounding medium, occurs due to the lag between the magnetization of the particles and the applied magnetic field. The efficiency of the particles is usually expressed as specific loss power (SLP), which is the mass normalized by observed power loss (W/mass) in the MNPs. In breast cancer, MNPs can be designed to play a multifunctional role including contrast agents for different imaging modalities for primary and metastatic breast cancer and for finding lymph nodes in sentinel lymph biopsy (SLNB), as carriers for drug delivery and as heating sources for magnetic particle hyperthermia [22].

MPH has long been studied since the seminal work by Gilchrist et al. in 1957 [23] and the feasible results of MPH for breast cancer treatment have already been proven. The studies reported in [18,19,24,25,26,27] showed similar results of reaching heating temperatures above 43 °C at tumor domains in order to cause coagulation necrosis and inhibit tumor growth. However, MPH has not yet been effectively used as a clinical routine worldwide due to several challenges such as the heating efficiency and biocompatibility of the MNPs, the delivery and localization of MNPs within the tissues, monitoring the temperature during treatment sessions and providing strong radiofrequency magnetic fields in large volumes while considering the biological safety limit. Particularly, exceeding the safety limit, which was originally [28] defined based on the amplitude (H) and frequency (f) of the applied field as H.f ≤ 4.85 × 10^8^ Am^−1^ s^−1^, can cause side effects in patients due to the induced eddy current. Although, Hergt et al. [29] showed that for small radii parts, this value can be one order greater (5 × 10^9^ Am^−1^s^−1^), an alternative method such as using a pulsed power can be utilized to reduce the induced eddy current. The pulsed power is generated when MNPs are exposed to an external alternating magnetic field (AMF) with a combination of different duties and cycles. In [30,31], the authors showed that the combination of high-amplitude AMF (>500 Oe) with pulsed power for a sufficient time led to tumor growth decay and did not cause toxicity to healthy tissue; while duties exceeding 50% resulted in occurrence of injuries in the mice. Temperature was regulated by varying the pulse for a specific amplitude and duty combination and finally the choice of the duty and cycle at amplitudes greater than 700 Oe was considered to prevent damage to the healthy tissues. Similarly, the optimum condition of intermittent field was also considered in [32]. The duty cycle of 33% with the ON/OFF condition of 50/100 in second for the intermittent magnetic field mode was shown to mitigate the eddy current in normal tissues. However, in the above results, the pulsed powers were only studied for reducing eddy current within a small temperature window of 41–45 °C, and the comparison, the optimized condition for duty and cycle of pulsed powers at different injection domains, and the explanation of the mechanism of tissue damage under were not investigated in detail.

In the present study, the effects of using pulsed powers of various duties and cycles with MNPs on the fraction of damage for thermal ablation of breast cancer were simulated and explained using COMSOL software and based on Fourier’s law, the Arrhenius and Pennes bio-heat equations. The pulsed powers with high duties (0.8 or 0.9) led to maximum treatment efficiency, even higher than continuous power at the small source domains (not larger than 4 mm). In addition, the effect of the cycles of the pulsed powers on heating efficiency was shown to be dependent on the source diameter. For treating larger source diameters, the fraction of damage in the source domain from continuous power was almost the same (only ~0.05% higher), but in healthy tissues was two times higher than that of the pulsed powers with high duties. These results may shed light on better understanding the effect of temperature fluctuation on tissue damage in MPH and allow better selection of a suitable applied magnetic field for treating breast cancer. In the end, we suggest solutions for MPH that not only may minimize the injury to the heathy tissues, but can also decrease the power required for the heating system.

## 2. Results

### 2.1. Determination of the Heating Power Necessary for MNPs in the Presence of Blood Flow and Metabolism to Effectively Damage the Tumor Tissues

From Equation (9), it is straightforward to estimate the MNPs heating power (P_0_) necessary to increase the temperature of the treated domain to 6 °C above the normal body temperature (37 °C) with the source diameter from 1 to 6 mm, 0.48 (W/(m.°C)) as thermal conductivity of the tumor, and in the absence of blood flow. From this heating power, the heating power necessary for MNPs in the presence of blood flow and metabolism (P_MNPs_) to reach the fraction of tumor damage of at least 70% was determined using the “parameter sweep” function in COMSOL. This fraction of damage is considerably higher than the fraction assumed in [33], which was 63% for tissue coagulation. The detailed information is shown in Table 1.

### 2.2. Determination of Optimized Pulsed Powers for Various Source Diameters to Get High Fraction of Tumor Damage

Figure 1 compares the fractions of tissue damage between the pulsed and continuous powers at source domains with different source diameters. For the source diameters up to 4 mm, the maximum fraction of damage was observed when using the pulsed powers with higher duties. By increasing the source diameter, this maximum fraction of damage for pulsed powers with high duty also shifted to the higher cycles. Specifically, the pulsed power with the duty of 0.8 and cycle of 5 s reached the highest fraction of tissue damage at source diameter of 1 mm. When increasing the source diameter from 2 to 6 mm, the duty of 0.9 worked best among the pulsed powers to achieve the higher fraction of tissue damage, while the cycle jumped from 10 s for the source diameter of 2 mm to 95 s for the source diameter of 6 mm. However, for source diameters larger than 4 mm, the continuous power achieved nearly the same fraction of tissue damage as the pulsed powers (~0.05% higher). The results of the maximum fraction of tissue damage among the pulsed powers and the results related to the continuous power are summarized in Table 2.

In order to damage not only the source domain, but also the entire tumor, we increased the heating power of the source for all diameters. Although for all source diameters the fraction of damage from continuous power was slightly higher (~6%) than that of the pulsed powers with high duty at the tumor domain, the fraction of tissue damage in tumor neighboring domain (healthy tissues) from the continuous power was two times higher than that of the pulsed powers. These can clearly be seen in Figure 2, which presents a typical result for source diameter of 2 mm. The detailed results can be found in Appendix A in the Appendix A. These results can be explained by the Arrhenius equation.

As it can be seen in Equations (11), (13) and (15) the fraction of tissue damage depends on the temperature exponentially. This means that a small change in the temperature can have a considerable effect on the fraction of tissue damage. Moreover, according to Equation (14), the slope of tissue damage is also exponentially related to the change in temperature. This slope represents the damage speed of living cells. A higher temperature results in an increase in the damage speed since molecules in cells have more energy to overcome the energy barrier E_a_ necessary to destroy the tissue. In other words, the temperature should be both high and fluctuated enough to increase the damage speed and rise the fraction of tissue damage. When the temperature from pulsed power decreases (ΔT < 0), the slope of tissue damage (k_2_ in Equation (12)) becomes almost zero and the fraction of tissue injury becomes almost constant ($k=\frac{\partial \Omega }{\partial t}$) according to Equations (11) and (14). However, increasing the temperature (ΔT > 0) leads to a significant increase in the fraction of tissue damage. On the other hand, the gradual increase in temperature caused by the application of continuous power (especially when reaching the saturation temperature) results in a low and almost constant slope, which in turn means a slow increase in the fraction of tissue damage. To make it clear, let us compare the fraction of tissue damage between the pulsed and continuous powers at source diameter of 1 mm, shown in Figure 3. The fraction of tissue damage from the pulsed power with cycle of 5 s and duty of 0.8 is higher due to achieving both higher average temperatures and more fluctuation in the temperature. The fraction of tissue damage from the pulsed power with cycle of 90 s and duty of 0.6 was the lowest among all the heating powers during the 30 min due to the remarkable decrease in temperature when the field is in the OFF state.

To explain the reason for the slightly better results observed for continuous power at larger source (>4 mm) or tumor domain (all diameters), let us see, as an example, the relationship between the temperature and the tissue damage of source domain at source diameter of 6 mm (Figure 4). The highest damage fraction among pulsed powers belongs to the duty of 0.9 and cycle of 95 s, which is still lower than continuous power. It is because of the lower temperature achieved by the pulsed power; however, when the temperature was high enough, at around 70 °C, the fraction of damage caused by the pulsed power was approximately equal to that of the continuous power. In addition, the fraction of damage under the pulsed power with a duty of 0.9 and cycle of 10 s was always lower than the others because of the lower achieved maximum temperature and the small temperature fluctuation.

When using the higher powers and reaching temperatures as high as 70 °C, one may consider these temperatures significantly higher than what a patient can withstand in real clinical hyperthermia applications. However, it should be noted that this is the average temperature inside the heating source which even at its maximum diameter (6 mm) is still more than 3 times smaller than the tumor diameter. Accordingly, the average temperature observed outside the heating source but within the tumor region was 44 °C (Figure 5a). In addition, the average temperature outside the tumor domain (heathy tissues) was observed to be 39.2 °C (Figure 5b) for pulsed power of 0.9 duty. Based on these results, reaching higher temperatures of even up to 70 °C at the source domain seems reasonable.

## 3. Discussion

According to our observations, the pulsed powers achieved either higher or almost the same fractions of damage as the continuous power for all diameters under the same inputs at the source (see Table 2). However, it may be more efficient to utilize pulsed powers for small diameter breast tumors (not larger than 4 mm) because of requiring a smaller heating power and a simpler magnetic field generation system. In addition, using pulsed powers for damaging the entire tumor leads to reducing the side effect of the eddy current to the surrounding heathy tissue for all conditions compared to continuous power—the method that is currently employed in typical magnetic hyperthermia therapy. It means that pulsed power can be a suitable choice for treating small breast tumors in terms of reducing the necessary heating power (the heating efficiency of the MNPs plus the applied magnetic field) and minimizing the side effects to the healthy tissues. It should, however, be noted that since the side effects of the pulsed powers are also less for larger source diameters, if a proper magnetic field generation system can be designed, utilizing the pulsed powers would be again more beneficial as compared to continuous power.

However, designing a system to apply a high pulsed power for human scale will be very challenging. One possible solution to apply the pulsed powers for magnetic hyperthermia can be the use of a combined magnetic particle imaging and magnetic hyperthermia system [34,35]. In this type of approach, MNPs can be injected into the region of interest (the entire tumor), which is aimed to be treated. The heating system can generate and move a field free region (FFR) at the targeted domain and only the MNPs located in the FFR can be excited by the pulsed magnetic field to generate the heat. Then, this targeted heating domain can be moved within the entire tumor. The advantage of this approach is that since the field is focused in small regions (i.e., small source domains), lower heating powers can be effectively used to damage them. Particularly, as shown above, heating small regions with high-duty magnetic fields significantly decreases the damage to the healthy tissues and hence reducing the side effects of magnetic hyperthermia therapy.

In this simulation study, it was assumed that the amount of MNPs used is enough to act as heating source for destroying the breast cancer tissue, but in reality, the delivery method of MNPs into the tumor should be considered to reach the expected dose of MNPs at tumor site for achieving the highest heating efficiency possible. MNPs can be delivered to the target tumors by intratumoral or intravenous injection [36,37]. Direct intratumoral injection leads to high concentration of MNPs at the tumor site and prevents systemic toxicity, but the tumors are not completely filled by MNPs. It is an invasive method and is applicable only for large tumors and not small metastatic tumor growths [37]. Although MNPs distribution in intravenous injection would not result in homogeneous distribution at the tumor site; it can be more global and systematic. This can evenly deliver the desired heating dose to the whole cancerous tissues even for regions having low concentration of MNPs [37]. For example, in [37], intravenous injection of 1.7 g Fe/kg weight of the animal led to the final concentration of 1.9 mg Fe/cc at the tumor site, which was enough for heating the tumor. To enhance the delivery of particles to the tumor site through intravenous injection, additional techniques such as magnetic particle imaging [38], swarm control in blood vessels [39] or utilizing a half of static saddle potential energy configuration for electromagnetic actuation system [40] combined with targeted drug delivery can be employed.

Moreover, the heating efficiency of MNPs exposed to AMF at targeted cells can also be limited due to tumor properties. The diverse tumor properties such as its heterogeneity and diameter can significantly affect the MNPs concentration, their heating efficiency, and the temperature distribution [41,42,43]. In addition, design and construction of devices that are capable of generating pulsed power with short cycles such as 10 s and duty of 0.9 is a challenging task. Advanced electronic circuit technology is required to achieve a highly exact pulsed magnetic field. Last but not least, the technology for non-invasively measuring and monitoring the temperature during the treatment process to avoid unexpected heating in healthy tissues is another challenge in magnetic hyperthermia. Some techniques such as magnetic [44,45] and ultrasound [46] based methods have already been proposed as potential candidates for this purpose.

## 4. Material and Methods

### 4.1. Breast Cancer Model and Bio-Heat Transfer Model

To study the fraction of tissue damage over time on cancerous breast tissue caused by the MNPs heat source while considering the heat losses from thermal conduction and convection (blood perfusion), the breast tissue model was simplified by assuming that the temperature distribution is homogeneous in the same layer. Figure 6 shows the hemispherical shaped breast tissue model with six layers [47]: epidermis (outer layer), papillary dermis, reticular dermis, fat, gland and, finally, muscle in the inner layer. The breast tumor was considered as a solid hemisphere of 20 mm diameter (considering as the stage II of breast cancer) centered 23 mm away from the epidermis layer. The heating source (MNPs) was assumed to be homogenously distributed in a solid hemisphere concentric with the breast tumor and with diameters changing from 1 to 6 mm. These sizes were chosen because the heating power of source diameters less than 1 mm was not enough to destroy the tumor and of the diameter larger than 5 mm showed the same optimized results. The physical properties of each tissue and the breast tumor are presented in Table 3.

Three different domains were defined to observe the fraction of tumor damage in each of them under application of the alternating magnetic field: (a) source domain—a volume containing injected nanoparticles for generating heat located in the center of the breast tumor, (b) tumor domain and (c) tumor neighboring domain—a part of the gland layer surrounding the breast tumor. The tumor neighboring domain is considered to evaluate the effect of the heating on healthy tissue.

The Pennes bio-heat transfer equation is a well-known thermal model for studying transient temperatures in a biological system. The equation evaluates heat transfer of metabolic processes and blood flow in living cells. The Pennes bio-heat transfer equation for breast model can be written as follows [47,53,54,55]:(1)$$\rho_{\mathrm{gl}}C_{\mathrm{gl}}\frac{\partial T}{\partial t}=k_{\mathrm{gl}}\nabla^{2}T+\rho_{b}C_{b}\omega_{b\_\mathrm{gl}}(T_{b}-T_{gl})+Q_{m\_gl}+f(t).P_{\mathrm{MNPs}}$$
(2)$$\rho_{n}C_{n}\frac{\partial T}{\partial t}=k_{n}\nabla^{2}T+\rho_{b}C_{b}\omega_{b\_n}(T_{b}-T_{n})+Q_{m\_n}$$

Here, ρ_b_, C_b_, T_b_ and ω_b_ are the density, specific heat, arterial temperature and perfusion rate of the blood, respectively. ρ_gl_, C_gl_, T_gl_, ω_b_gl_, k_gl_ and Q_m_gl_ are, respectively, the density, specific heat, temperature, blood perfusion rate, thermal conductivity and metabolic generation of the gland. r_n_, C_n_, T_n_, ω_b_n_, k_n_ and Q_m_n_ for n = 1 to 5 are the density, specific heat, temperature, blood perfusion rate, thermal conductivity and metabolic generation of the remaining layers (epidermis, papillary dermis, reticular dermis, fat and muscle) of the breast model, respectively. P_MNPs_ is the volumetric power dissipation of the MNPs under AMF (W/m^3^). Finally, f(t) is the pulsed function to control the pulsed and continuous heating powers with different cycles and duties defined by the following equation.
(3)$$\mathrm{duty}=\frac{T_{\mathrm{on}}}{T_{\mathrm{on}}+T_{\mathrm{off}}}=\frac{T_{\mathrm{on}}}{\mathrm{cycle}}$$

Here, T_on_ and T_off_ are the times for activating and deactivating the heating power during one period and “cycle” is the time for one period. Then, duty = 1.0 represent the continuous power.

As mentioned earlier, air temperature distribution was assumed to be uniform for heat exchange with the outer layer (epidermis) and there was no heat loss between the layers inside the breast model during the heat exchange process. To solve Equations (1) and (2), the generalized boundary conditions for heat flux are necessary.

Heat flux and temperature continuity between layers inside the breast model are:(4)$$k_{n-1}\frac{\partial T_{n-1}}{\partial \eta }=k_{n}\frac{\partial T_{n}}{\partial \eta }$$
(5)$$T_{0\_n-1}=T_{0\_n}$$
where η is perpendicular to the surface and T_0_n_ is the initial temperature of the n^th^ layer.

The boundary condition for heat exchange between the epidermis layer T_1_ and the surrounding air T_air_ is:(6)$$k_{1}\frac{\partial T_{1}}{\partial \eta }=-h(T_{1}-T_{\mathrm{air}})$$
where the heat transfer coefficient h = 10 W/(m^2^.K) and T_air_ = 25 °C.

The initial temperature T_0_n_ of the layers was considered as:(7)$$T_{0\_n}=37{}^{\circ}C,t=0$$

The body core temperature T_c_ was considered as a constant to maintain the body temperature of almost 37 °C at the boundary between the core body and the innermost layer (muscle):(8)$$T_{5}=T_{c}=37{}^{\circ}C$$

### 4.2. Fourier’s Law for Estimating Heating Power Necessary for Magnetic Nanoparticles

Based on the dimensionless Fourier’s law for thermal analysis of nanoparticles in biological material, the volumetric power dissipation (P_0_) for MNPs that is necessary to increase the temperature (ΔT) in the steady-state condition can be estimated as [56]:(9)$$P_{0}=\frac{8k*\Delta T}{d_{}^{2}}$$
where d (m) is the diameter of the injected region and k (W/m.K) is the thermal conductivity of the biological tissues.

It should be noted that only the heating power necessary for nanoparticles in the absence of blood flow and metabolism can be estimated from Fourier’s law.

### 4.3. Arrhenius Equation for Fraction of Tissue Damage

In physical chemistry, the Arrhenius equation describes the relationship between temperature and chemical reaction rate. This equation plays an important role in determining the rate of a chemical reaction and the activation energy of colliding molecules upon changing the temperature. The Arrhenius equation is of the form [57,58]:(10)$$k=Ae^{\frac{-E_{a}}{RT_{}}}$$
where k is the rate of molecules in the chemical reaction, A is the colliding frequency factor or pre-exponential factor (1/s), T is the temperature (K), E_a_ is the activation energy (J/mol) and R = 8.3145 J/(mol.K) is the universal gas constant. The value of “A” relates to the frequency of collisions and the orientation of a favorable collision probability while E_a_ is the energy barrier to form the transition state of the critical target during the rate-limiting step of inactivation. The parameters k and A are independent of temperature and identified experimentally. Here, the A and E_a_ parameters for breast cancer are 1.18 × 10^44^ 1/s and 3.02 × 10^5^ J/mol, respectively [58].

In the breast cancer model, Equation (10) can be rewritten as:(11)$$k=Ae^{\frac{-E_{a}}{RT_{n}}}$$

The Arrhenius equation shows that the reaction rate k has an exponential relationship with E_a_ and T_n_. To observe the effect of temperature variation on reaction rate, Equation (11) can be rewritten for two different temperatures as:(12)$$\begin{array}{c} \begin{array}{c} \ln k_{1}=\ln A-\frac{E_{a}}{RT_{n1}}\\ \ln k_{2}=\ln A-\frac{E_{a}}{RT_{n2}} \end{array}\}\Rightarrow \ln k_{2}-\ln k_{1}=\ln\frac{k_{2}}{k_{1}}=\frac{E_{a}}{R}\left[\frac{T_{n2}-T_{n1}}{T_{n1}.T_{n2}}\right]\\ \Rightarrow k_{2}=k_{1}e^{\frac{E_{a}}{R}\left[\frac{\Delta T_{n}}{T_{n1}.T_{n2}}\right]} \end{array}$$
where k_1_ and k_2_ are the rates of the chemical reaction at temperatures T_n1_ and T_n2_ at two different times, respectively. Thus, a higher temperature difference ΔT_n_ will lead to a higher k_2_ value as compared to k_1_.

In thermal ablation, the dependence of the degree of tissue injury Ω on the temperature and treatment time is described using the Arrhenius equation as [58,59]:(13)$$\Omega =\int_{0}^{t}Ae^{\frac{-E_{a}}{RT_{n}}}dt$$
(14)$$\Rightarrow \frac{\partial \Omega }{\partial t}=Ae^{\frac{-E_{a}}{RT_{n}}}$$

Then, the fraction of tissue damage, θ, is shown in relation with degree of tissue injury as [58]:(15)$$\theta =1-e^{-\Omega }$$

From Equations (11)–(15), it is apparent that the fluctuation in temperature of a tissue at two different times leads to change in the reaction rate in an exponential form, which is in fact the change of the speed or slope of the degree of tissue injury (damage). The results in [33] showed that a fraction of damage exceeding 63% causes tissue coagulation in living cells.

### 4.4. Methods

In this study, COMSOL software was used to build the breast cancer model shown in Figure 6 with the parameters for each layer set as presented in Table 3. As mentioned earlier, the heat sources were solid hemisphere concentric with the breast tumor with diameters ranging from 1 to 6 mm. These sources were assumed to be filled homogenously with MNPs, which are the heat mediators for magnetic hyperthermia. All these MNPs were assumed to have the same heating efficiency (or volumetric power dissipation). The concentration of the MNPs in the heating sources were also kept constant.

The process of evaluating the optimized condition of the pulsed powers based on the fraction of tissue damage in the breast tumor model is presented in Figure 7. The heating power (P_0_) necessary for MNPs to reach the desired temperature in the absence of blood flow can be estimated by Fourier’s law (Equation (9)). Using the “parameter sweep” function in COMSOL software, we can estimate the heating power (P_MNPs_) for nanoparticles to destroy the targeted domain under the presence of blood flow and metabolism to get at least 70% of fraction of tissue damage. According to this heating power, various pulsed powers with different combinations of cycles and duties, which are defined in Equation (3), were applied to the breast model to obtain the optimized condition of the pulsed power for reaching the maximum fraction of tissue damage. The process was repeated for different source domain diameters which can also be considered as changing the injected particle volume and the heating power. The results were explained based on the Arrhenius equation. All simulations were performed for 30 min, similar to several in vivo reports [18,19,26,27].

## 5. Conclusions

In thermal ablation, destroying the tumor is the final target but its effect on the healthy tissue also needs to be considered. The pulsed powers are utilized to minimize this effect while still achieving a high fraction of cancerous tissue damage. Based on our simulation study, the optimized conditions of the pulsed powers at various source diameters were determined. The pulsed powers with high duty (0.8 and 0.9) showed better performance in terms of treatment efficiency when the source diameter was increased from 1 to 6 mm, which was even higher than that for the continuous power for source diameters not larger than 4 mm. The effect of the cycle of the pulsed power on its efficiency also depends on the source diameter, where the larger source diameter is, the longer cycle should be utilized. Moreover, the use of pulsed power for tumor diameters not larger than 4 mm can reduce the cost due to the use of a simpler heating system and minimizing the side effects in the healthy tissues. Depending on the stage of cancer or tumor diameter, the heating power system and the status of the patients, one can choose the suitable condition of duty and cycle for pulsed powers in clinical trials. The application of pulsed powers in a focused heating system to get higher efficiency of tumor damage can be the subject of future experimental studies.

## Figures and Tables

**Figure 1 ijms-22-08895-f001:**
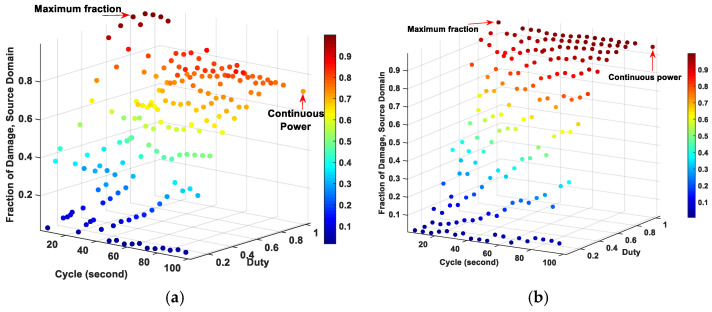
Fraction of tissue damage at source domain by changing source diameter: (**a**) 1 mm, (**b**) 2 mm, (**c**) 3 mm, (**d**) 4 mm, (**e**) 5 mm and (**f**) 6 mm.

**Figure 2 ijms-22-08895-f002:**
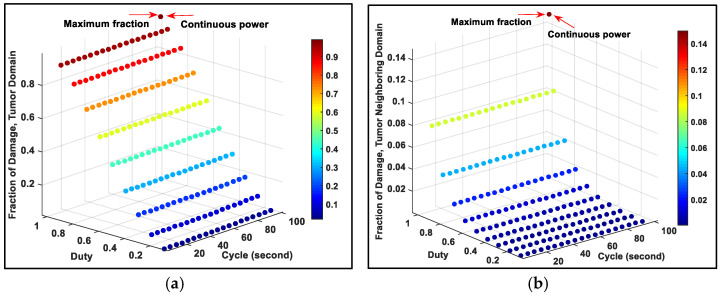
Fraction of tissue damage at (**a**) tumor domain and (**b**) tumor neighboring domain for source diameter of 2 mm for pulsed and continuous powers for 30-min simulations.

**Figure 3 ijms-22-08895-f003:**
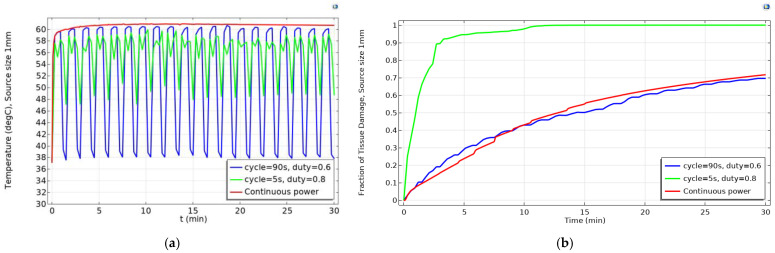
The change in (**a**) the temperature and (**b**) the fraction of tissue damage at the source diameter of 1 mm with pulsed powers during the 30-min simulation.

**Figure 4 ijms-22-08895-f004:**
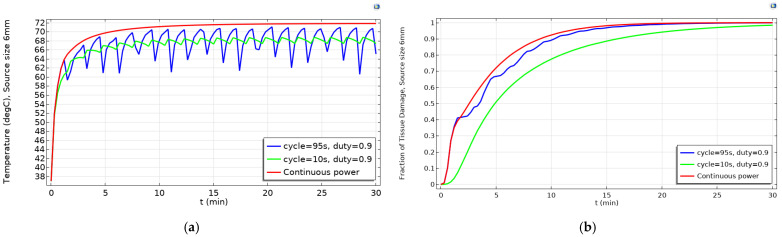
The change in (**a**) the temperature and (**b**) the fraction of tissue damage at the source diameter of 6 mm with pulsed powers during the 30-min simulation.

**Figure 5 ijms-22-08895-f005:**
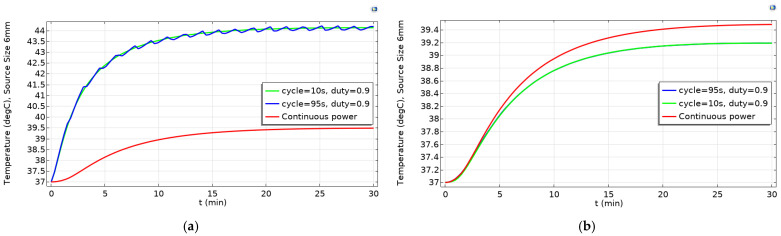
The temperature at (**a**) tumor domain and (**b**) tumor neighboring domain with source diameter of 6 mm during the 30-min simulation.

**Figure 6 ijms-22-08895-f006:**
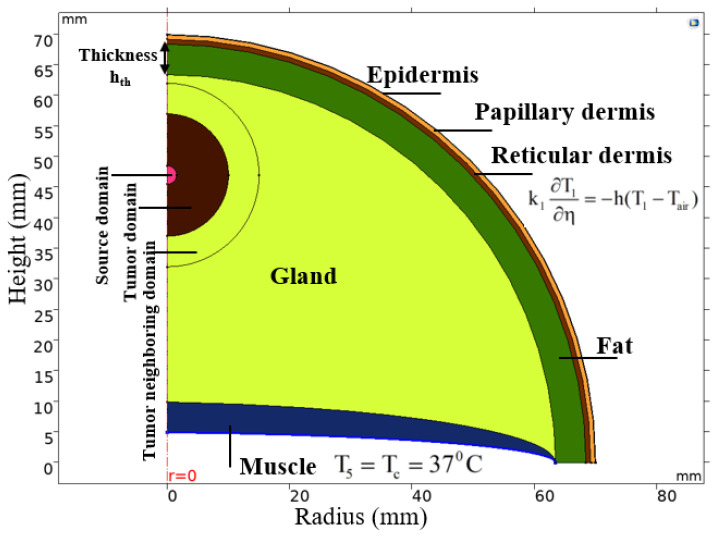
Multi-layered axisymmetric 2D breast model.

**Figure 7 ijms-22-08895-f007:**
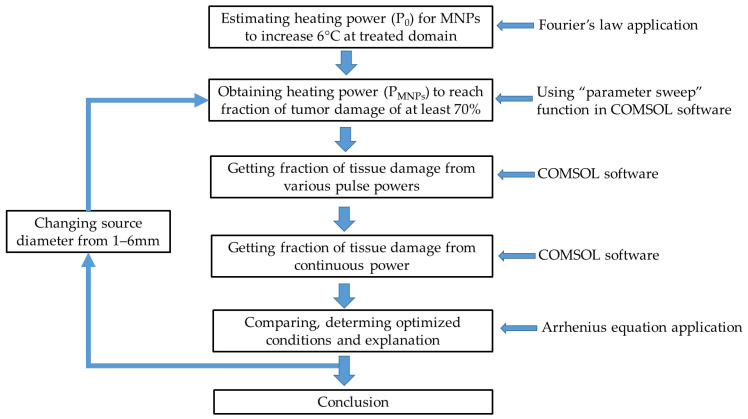
Flowchart of the methods and processes used in this simulation.

**Table 1 ijms-22-08895-t001:** Estimating the heating powers necessary for MNPs to reach fraction of tumor damage of at least 70%.

Diameter of Source Domain (mm)	P_0_ (W/m^3^) to Increase 6 °C at Treated Domain	P_MNPs_ (W/m^3^) to Reach Fraction of Tumor Damage of at Least 70%
1	2.30 × 10^7^	4.8 × 10^7^
2	0.57 × 10^7^	3.6 × 10^7^
3	0.25 × 10^7^	2.2 × 10^7^
4	0.14 × 10^7^	1.9 × 10^7^
5	0.92 × 10^7^	1.8 × 10^7^
6	0.64 × 10^7^	1.7 × 10^7^

**Table 2 ijms-22-08895-t002:** Summary of the maximum fraction of tissue damage among the pulsed powers and fraction of tissue damage of the continuous power at the source domain.

Source Diameter (mm)	Pulsed Powers		Fraction of Damage of Pulsed Power	Fraction of Damage of Continuous Power
	Duty	Cycle (second)		
1	0.8	5	0.99983	0.71695
2	0.9	10	0.99627	0.9543
3	0.9	90	0.93884	0.89937
4	0.9	90	0.97700	0.96749
5	0.9	95	0.99130	0.99213
6	0.9	95	0.99903	0.99972

**Table 3 ijms-22-08895-t003:** The physical properties of each tissue and the breast tumor.

	Epidermis	Papillary Dermis	Reticular Dermis	Fat	Gland	Muscle	Tumor	Air
Thickness h_th_ (mm)	0.1 [48]	0.7 [48]	0.8 [48]	5.0 [49]	43.4 [49]	15 [49]	20	--
k (W/(m.K))	0.235 [48]	0.445 [48]	0.445 [48]	0.21 [50]	0.48 [50]	0.48 [50]	0.48 [50]	--
ρ (kg/m^3^)	1200 [48]	1200 [48]	1200 [48]	930 [51]	1050 [51]	1100 [51]	1050 [51]	--
c (J/(kg.K))	3589 [48]	3300 [48]	3300 [48]	2770 [51]	3770 [51]	3800 [48]	3852 [48]	--
Q_0_ (W/m^3^)	0 [48]	368.1 [48]	368.1 [48]	400 [50]	700 [50]	700 [50]	5000 [52]	--
ω_b_ (m^3^/((s.m^3^))	0 [48]	0.0002 [48]	0.0013 [48]	0.0002 [47]	0.0006 [47]	0.0009 [47]	0.012 [47]	--
Initial temperature T_0_ (°C)	37	37	37	37	37	37	37	25

## Data Availability

The data presented in this study are available in Appendix A.

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
