# Peer review of "Theoretical Analysis for Using Pulsed Heating Power in Magnetic Hyperthermia Therapy of Breast Cancer"

_ijms, 2021, doi:10.3390/ijms22168895_

Round 1
Reviewer 1 Report
In this manuscript, authors used a theoretical analysis to promote the magnetic hyperthermia therapy of breast cancer with treatment of pulsed heating power. The calculated results showed that using pulsed heating power could get better efficiency than using continuous power. These simulation results could provide valuable information to treat breast cancer with the hyperthermia therapy. At the same time, some issues also needed to be concerned. Therefore, I recommend this manuscript can be accepted after minor revision.
I have some questions as below.
- In this manuscript, authors wanted to use magnetic nanoparticle (MNP) as a heat-generated source. But authors used particles with diameters between 1~6 mm as magnetic materials. I think they cannot define as "nanoparticle", they were more bigger than nanoparticle. They should be micro-sized particles. Authors should re-modify the word "nanoparticle" in this manuscript.
- In different sized magnetic particles, were their Fe ion concentrations the same in all sized particle in this simulation? If yes, it means the particle number of different sized magnetic particles were different. The magnetic particles with 1 and 6 mm diameter respectively had the most and fewest particles. The particle number of magnetic materials is an important parameter that controls the heat-generated efficiency. Authors should consider this in this manuscript.
Author Response
Point 1: In this manuscript, authors wanted to use magnetic nanoparticle (MNP) as a heat-generated source. But authors used particles with diameters between 1~6 mm as magnetic materials. I think they cannot define as "nanoparticle", they were more bigger than nanoparticle. They should be micro-sized particles. Authors should re-modify the word "nanoparticle" in this manuscript.
Response 1: Thank you for the precise attention to the mistake in the methods section. In fact, the sentence in lines 232–233 (In this study, as mentioned earlier, it was assumed that MNPs as the source having different diameters (1 to 6 mm) are injected at the center of the breast tumor) is a grammatical error by the authors that must be corrected. Recalling section 2.1, where we have described the breast cancer model, can clearly show the error that has happened in lines 232–233. We have built the multi-layered model with a 2 cm diameter tumor, and heat sources with diameters from 1 to 6 mm are used to induce damage to the tumor. These sources are assumed to be filled with magnetic nanoparticles, which are the heat mediators for magnetic hyperthermia. We have modified this sentence in the revised manuscript in lines 231–233.
Point 2: In different sized magnetic particles, were their Fe ion concentrations the same in all sized particle in this simulation? If yes, it means the particle number of different sized magnetic particles were different. The magnetic particles with 1 and 6 mm diameter respectively had the most and fewest particles. The particle number of magnetic materials is an important parameter that controls the heat-generated efficiency. Authors should consider this in this manuscript.
Response 2: We agree with the reviewer and believe that the answer to point 1 clarifies this point to a great extent. However, to elaborate further, we have added more explanation to the new version of the manuscript. In our study, we have assumed that all injected magnetic nanoparticles have the same heating efficiency. The only difference here is the amount of the injected particles which is changed based on the heating source diameters. However, the concentration in all the sources was considered to be the same. A description about this has been added to the new version of the manuscript in lines 234–236.
Reviewer 2 Report
This manuscript contains a very interesting idea, which will probably show the way to other investigations, in which animal experimentation and in vitro (as such, of limited transcendence) cell viability tests can be ubstituted (at the initial stages of new treatments based on physical technologies) by sinulations. This is a significant merit of the manuscript. The text is well written and clear, and I have some minor considerations only:
a) In some places of the Introduction, it appears to be suggested that hyperthermia is a routine therapy, which, as far as I know, is not the case.
b) For readers unfamiliar with it, he Pennes model should receive more attention in its description: a few words on its basis, an explanation of each of the terms in eq. 1, at least
c) Fig. 2, explaining the meaning of Ton and Toff is unnecessary.
d) In many places Kelvin degrees are denoted oK, instead of K. Also, in several Figures cycle is written as cylce
e) Table 4: how can you establish the number of significant figures?
f) In temperature plots (Figs. 6, 7), values as high as 70 degrees are reached. Is this reasonable? It appears a largely destroying temperature for the tissue and surroundings. I do not know if a patient might withstand this (70 degrees in some regions of the body)
Author Response
The authors appreciate the constructive comments and suggestions from the reviewers. The manuscript has been revised based on these comments. The changes have been highlighted in the revised manuscript.
Point 1: In some places of the Introduction, it appears to be suggested that hyperthermia is a routine therapy, which, as far as I know, is not the case.
Response 1: Thank you for reviewer’s comment. We agree with this comment and sections related to this subject has been modified in the revised version. Please see lines 50,51 and 57.
Point 2: For readers unfamiliar with it, the Pennes model should receive more attention in its description: a few words on its basis, an explanation of each of the terms in eq. 1, at least.
Response 2: Based on the reviewer’s comment, we added the explanation about the Pennes bio-heat equation. Please check lines 145–147 in Section 2.1.
Point 3: Fig. 2, explaining the meaning of Ton and Toff is unnecessary.
Response 3: Thank you for the reviewer’s comment. Modifications related to this comment have been applied to the new version of the manuscript.
Point 4: In many places Kelvin degrees are denoted oK, instead of K. Also, in several Figures cycle is written as cylce.
Response 4: Thank you for the reviewer’s comments. We have corrected the errors.
Point 5: Table 4: how can you establish the number of significant figures?
Response 5: We agree with reviewer’s comment and the table 4 appears to be a bit confusing. In fact, we have tried to summarize the results of maximum tissue damage that was observed for each heating source to compare with the continuous power. Please see the changes in the table and its description.
Point 6: In temperature plots (Figs. 6, 7), values as high as 70 degrees are reached. Is this reasonable? It appears a largely destroying temperature for the tissue and surroundings. I do not know if a patient might withstand this (70 degrees in some regions of the body)
Response 6: As properly mentioned by the reviewer, withstanding temperatures around 70 degrees does not seem to be feasible or reasonable. However, we should notice that the temperature 70 degrees reported in our study is the average temperature inside the heating source which even at its maximum diameter (6 mm) is more than 3 times smaller than the tumor diameter. Accordingly, the observed average temperature outside the heating source but within the tumor region was 44 degrees (figure 7.a). In addition, the temperature outside the tumor domain (heathy tissues) was observed to be 39.2 (figure 7.b) for pulsed power of 0.9 duty. Based on these results, reaching higher temperatures of even up to 70 degrees at source domain seems to be reasonable. These explanation has been added to the new version of the manuscript.